

# *In silico* analysis reveals a shared immune signature in *CASP8*-mutated carcinomas with varying correlations to prognosis

Yashoda Ghanekar[1,*] and Subhashini Sadasivam[1,2,*]

[1] DeepSeeq Bioinformatics, Bangalore, Karnataka, India
[2] Institute for Stem Cell Biology and Regenerative Medicine, Bangalore, Karnataka, India
[*] These authors contributed equally to this work.

## ABSTRACT

**Background.** Sequencing studies across multiple cancers continue to reveal mutations and genes involved in the pathobiology of these cancers. Exome sequencing of oral cancers, a subset of Head and Neck Squamous cell Carcinomas (HNSCs) common among tobacco-chewing populations, revealed that ∼34% of the affected patients harbor mutations in the *CASP8* gene. Uterine Corpus Endometrial Carcinoma (UCEC) is another cancer where ∼10% cases harbor *CASP8* mutations. Caspase-8, the protease encoded by *CASP8* gene, plays a dual role in programmed cell death, which in turn has an important role in tumor cell death and drug resistance. CASP8 is a protease required for the extrinsic pathway of apoptosis and is also a negative regulator of necroptosis. Using multiple tools such as differential gene expression, gene set enrichment, gene ontology, *in silico* immune cell estimates, and survival analyses to mine data in The Cancer Genome Atlas, we compared the molecular features and survival of these carcinomas with and without *CASP8* mutations.

**Results.** Differential gene expression followed by gene set enrichment analysis showed that HNSCs with *CASP8* mutations displayed a prominent signature of genes involved in immune response and inflammation. Analysis of abundance estimates of immune cells in these tumors further revealed that mutant-*CASP8* HNSCs were rich in immune cell infiltrates. However, in contrast to Human Papilloma Virus-positive HNSCs that also exhibit high immune cell infiltration, which in turn is correlated with better overall survival, HNSC patients with mutant-*CASP8* tumors did not display any survival advantage. Similar analyses of UCECs revealed that while UCECs with *CASP8* mutations also displayed an immune signature, they had better overall survival, in contrast to the HNSC scenario. There was also a significant up-regulation of neutrophils ($p$-value = 0.0001638) as well as high levels of IL33 mRNA ($p$-value = 7.63747E−08) in mutant-*CASP8* HNSCs, which were not observed in mutant-*CASP8* UCECs.

**Conclusions.** These results suggested that carcinomas with mutant *CASP8* have broadly similar immune signatures albeit with different effects on survival. We hypothesize that subtle tissue-dependent differences could influence survival by modifying the micro-environment of mutant-*CASP8* carcinomas. High neutrophil numbers, a well-known negative prognosticator in HNSCs, and/or high IL33 levels may be some of the factors affecting survival of mutant-*CASP8* cases.

Corresponding authors
Yashoda Ghanekar,
yashoda.ghanekar@gmail.com
Subhashini Sadasivam,
contact@deepseeq.com,
deepseeq@gmail.com

## INTRODUCTION

Exome sequencing, RNA-sequencing, and copy number variation analysis of different cancers have revealed a cornucopia of disease-relevant mutations and altered pathways (*Cancer Genome Atlas Research Network et al., 2013b*). The identified genes included those with broad relevance across different cancers, as well as those relevant in one or few cancer types. The next phase will involve parsing this voluminous data to generate ideas and hypotheses with the potential for clinical impact, and then testing them experimentally.

We are particularly interested in the heterogeneous group of Head and Neck Squamous cell Carcinomas (HNSCs) as these account for a large number of mortalities each year in the Indian subcontinent (*Ferlay et al., 2010*; *Gupta et al., 2011*). Multiple exome sequencing studies have revealed the landscape of recurrent somatic mutations in HNSCs and its prevalent subtype of Oral Squamous Cell Carcinomas (OSCCs) (*Agrawal et al., 2011*; *India Project Team of the International Cancer Genome Consortium, 2013*; *Pickering et al., 2013*; *Stransky et al., 2011*; *Hayes et al., 2016*). While *TP53* was the most significant recurrently mutated gene in this cancer type, several other genes such as *CASP8*, *FAT1*, and *NOTCH1* were also unearthed as significantly recurrently mutated by these large-scale sequencing studies. Barring *TP53*, the roles of these genes in oral epithelium homeostasis, and how this is altered owing to their mutation in cancer remain to be fully elucidated (*Rothenberg & Ellisen, 2012*). In this study, we chose to focus on the *CASP8* gene, which is mutated in ~10% of all HNSC cases, and more specifically in 34% of cases with OSCC of the gingiva-buccal sulcus (OSCC-GB), the subtype that accounts for the majority of HNSC cases in the Indian subcontinent (*Agrawal et al., 2011*; *Stransky et al., 2011*; *Hayes et al., 2016*). The types of mutations in *CASP8* reported in these HNSC cases included loss of function due to frameshift, nonsense mutation or splice mutation as well as missense and deletion mutations.

Apart from HNSC, Uterine Corpus Endometrial Carcinoma (UCECs) carried the most numbers of mutations in the *CASP8* gene, as was observed upon searching the Genomic Data Commons (*Grossman et al., 2016*). We found that *CASP8* was recurrently mutated in about 10% of UCEC cases. Here again, the role of *CASP8* in endometrial tissue homeostasis, and how this is altered owing to its mutation in UCEC remains unclear. *CASP8* was also mutated in other cancer types, however, the numbers of such tumors are too low for meaningful analyses. Thus, using the sequencing data on 528 head and neck, and 560 uterine corpus endometrial carcinoma tumors available in The Cancer Genome Atlas (TCGA) (*Cancer Genome Atlas Network, 2015*; *Cancer Genome Atlas Research Network et al., 2013a*), we sought to identify distinctive features of mutant-*CASP8* tumors.

CASP8 regulates two pathways of programmed cell death; it is a key protease required for the initiation of the extrinsic apoptotic pathway that is targeted by some drug-resistant tumors, and it is an important negative regulator of necroptosis (*Pasparakis & Vandenabeele, 2015*; *Feltham, Vince & Lawlor, 2017*; *Günther et al., 2011*; *Weinlich et al., 2013*). Loss-of-function mutations in *CASP8* could lead to reduced apoptosis and promote tumor survival (*Salvesen & Walsh, 2014*). It could also lead to enhanced necroptosis and

promote tumor cell death (*Günther et al., 2011*; *Weinlich et al., 2013*). Interestingly, it has been proposed that the necroptotic pathway could be utilized to develop anti-cancer treatments for countering cancers with resistance to apoptosis (*Su et al., 2016*). At least four HNSC-associated *CASP8* mutations have been reported to inhibit activation of the extrinsic apoptosis pathway suggesting loss-of-function, however necroptosis was not analyzed in this study (*Li et al., 2014*). On the background of these observations, tumors harboring *CASP8* mutations offer a tractable, physiologically relevant opportunity to understand the changes brought about by *CASP8* mutation, how it affects survival, and if *CASP8* or the necroptotic pathway could be a potential drug target.

In this study, we describe the comparison of RNA-sequencing (RNA-seq) data from head and neck squamous cell carcinoma, and later from uterine corpus endometrial carcinoma, that are mutant or wild type for *CASP8*. We report distinctive molecular features of mutant-*CASP8* HNSCs and UCECs that this comparison revealed. In addition, we describe results obtained by correlating these features to overall survival in the affected patients.

## MATERIALS AND METHODS

### Differential gene expression analysis of wild-type-*CASP8* and mutant-*CASP8* cases

Data for 528 head and neck squamous cell carcinoma (HNSC) cases available at The Cancer Genome Atlas (TCGA) were downloaded in May–June 2017 from https://portal.gdc.cancer.gov/. Clinical data files, Mutation Annotation Format (MAF) files, and mRNA quantification files such as HT-Seq files (files with number of reads aligning to each protein-coding gene) and FPKM-UQ files (files with number of fragments aligning per kilobase of transcript per million mapped reads normalized to upper quartile) were downloaded. The HPV status of HNSC cases at TCGA has been reported earlier (*Chakravarthy et al., 2016*), and these data were used to assign HPV-positive and HPV-negative cases.

Cases with and without *CASP8* mutation were selected as shown in Fig. 1. *CASP8* mutations in HNSC cases were identified using the Mutation Annotation Format (MAF) files available at TCGA. The workflow for somatic mutation calling at TCGA uses four different pipelines: SomaticSniper, MuSE, MuTect2, and VarScan2. The variants called by these four pipelines are further annotated to infer the biological context of each variant using Variant Effect Predictor (VEP). VEP predicts the effect of variants based on its location and information from databases such as GENCODE, sift, ESP, polyphen, dbSNP, Ensembl genebuild, Ensembl regbuild, HGMD and ClinVar. This annotation results in a list of variants with three predicted effects; high impact variants arising from frame-shift or nonsense mutations, variants with moderate impact which include missense mutations and low impact which include variants with synonymous mutations. The information regarding the impact of mutations was available in the MAF file from each somatic mutation calling pipeline employed by TCGA.

Fifty-five HNSC cases with non-synonymous *CASP8* mutations were identified from MAF files. Notably, the majority (80%) of the identified *CASP8* mutations were predicted

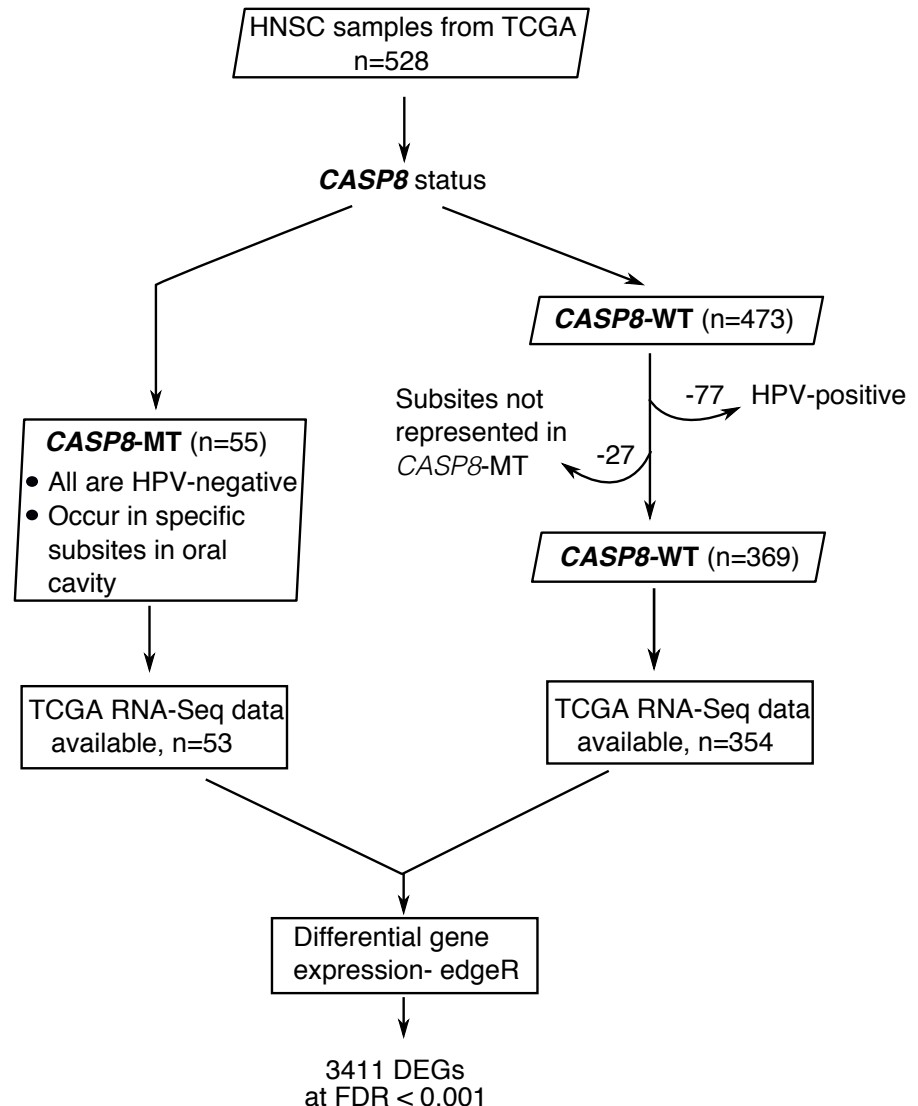

**Figure 1** **A flowchart indicating the sequence of processes used to select the HNSC cases used in this study.** HNSC cases with *CASP8* mutation were identified using MAF files from TCGA. Out of 528 HNSC cases available at TCGA, 55 cases had mutations in *CASP8*. All cases with *CASP8* mutation were HPV-negative. Hence, HPV-negative wild-type cases were considered for use as control. In addition, as *CASP8*-MT cases occurred in specific subsites in oral cavity (Table S1), *CASP8*-WT cases from these same subsites were selected as control. Thus, 369 HNSC cases with wild-type-*CASP8* were selected as control. Gene expression data was available for 53 cases with *CASP8* mutations and 354 cases with wild type *CASP8*. Data from HT-Seq files of selected cases with *CASP8* mutation and corresponding wild-type control cases was analyzed using edgeR to identify genes that were differentially expressed in *CASP8*-MT HNSCs as compared to *CASP8*-WT. *DEGs*, Differentially Expressed Genes; *FDR*, False Discovery Rate.

by more than one somatic mutation calling pipeline, and all had either high or moderate impact on function. All cases with *CASP8* mutation were HPV-negative and were found in tumors at specific sites in oral cavity. Therefore, HNSC cases that were from these same subsites and were HPV-negative were used as wild type control. A total of 424 HNSC cases

of which 369 had wild-type-*CASP8* (*CASP8*-WT) and 55 had mutant-*CASP8* (*CASP8*-MT) were thus selected (Table S1). Among selected cases, RNA-seq data was available for 354 cases with wild-type-*CASP8* and 53 cases with mutant-*CASP8*.

Transcripts that were differentially expressed in *CASP8*-MT as compared to *CASP8*-WT cases were identified using edgeR (*Robinson, McCarthy & Smyth, 2010*). edgeR uses raw read counts as input, which were obtained from HT-Seq files. The analysis was performed using quantile-adjusted conditional maximum likelihood (qCML) method without any filters. All transcripts with FDR < 0.001 and showing a fold-change of at least 2.5-fold (logFC of 1.3) were deemed to be significantly differentially expressed.

Similarly, clinical data and HT-Seq files for 560 uterine corpus endometrial carcinoma (UCEC) cases available at TCGA were downloaded in February 2018. *CASP8* mutations of high or moderate impact were present in 56 UCEC cases. Cases without *CASP8* mutations were used as wild type control. RNA-seq data was available for 476 *CASP8*-WT tumors and 56 *CASP8*-MT tumors. Transcripts that were differentially expressed in *CASP8*-MT as compared to *CASP8*-WT cases were identified using edgeR as described for differential gene expression analysis of HNSC.

## Gene Ontology and Gene Set Enrichment Analysis (GSEA)

Enrichment analysis was performed at http://geneontology.org/ to identify biological processes overrepresented among transcripts that were differentially expressed between *CASP8*-WT and *CASP8*-MT HNSCs (*The Gene Ontology Consortium, 2017*). Genes that passed the following criteria: (a) FDR < 0.001 (b) FDR < 0.001 and log2FC < −1.3, (c) FDR < 0.001 and log2FC > 1.3, (d) b and c merged, were used to create input gene sets for gene ontology analysis performed using PANTHER version 13.1 (release 2018-02-03). The Binomial test was used to determine statistical significance and the Bonferroni correction for multiple testing was applied.

GSEA was performed using a pre-ranked gene list and hallmark gene sets available at the Molecular Signature Database (*Subramanian et al., 2005*). The hallmark gene sets use either HGNC or entrez gene ids as the gene identifier. Out of the 60,483 transcripts analyzed by edgeR, HGNC gene symbols could be assigned to 36,095. logFC values for these 36,095 transcripts from the edgeR output from HNSC and UCEC differential gene expression analyses were used to generate the pre-ranked gene list. Gene sets with FDR < 25% and with distinct enrichment at the beginning or end of the ranked list (as observed in enrichment plots) were taken to be significantly enriched gene sets. To perform GSEA with genes that were up-regulated in the skin of mice lacking functional *Caspase-8*, the top 100 up-regulated genes that were reported were selected (*Kovalenko et al., 2009*). Of these 100 genes, 80 genes had corresponding human orthologs (*CASP8*-KOSET) as identified using tools available at http://www.informatics.jax.org/. GSEA was then performed using the pre-ranked gene list and the *CASP8*-KOSET.

## Immune cell infiltration in HNSC and UCEC cases

The abundance estimates of six immune cell types; B cells, CD4[+] T cells, CD8[+] T cells, Neutrophils, Macrophages, and Dendritic cells in TCGA cases was downloaded from

Tumor IMmune Estimation Resource (TIMER) at https://cistrome.shinyapps.io/timer/ (*Li et al., 2017*). This data was available for 353 *CASP8*-WT and 51 *CASP8*-MT cases from HNSC. The comparison of immune cell infiltration levels across *CASP8*-WT, *CASP8*-MT, and HPV-positive cases was performed using a two-sided Wilcoxon rank test and the graphs were plotted using R. Similar analysis was performed for all *CASP8*-MT and *CASP8*-WT cases from UCEC.

## Survival analysis

Survival analysis was performed to investigate the difference in the survival of *CASP8*-WT and *CASP8*-MT patients from HNSC and UCEC. Survival analysis was also performed to investigate the effect of factors such as expression levels of certain genes and immune cell infiltration on survival. The expression levels of genes of interest were obtained from FPKM-UQ files from TCGA and the data for distribution of immune cell infiltration was obtained from TIMER. Kaplan–Meier curves for *CASP8*-WT and *CASP8*-MT cases were plotted using the Survival and Survminer packages in R and the plots were compared using the log-rank test (*Therneau, 2015*).

To investigate the effect of genes of interest (such as those from gene sets enriched in GSEA or genes involved in necroptosis) and immune cell infiltration levels on survival, multivariate Cox proportional hazards test was performed for HNSC cases. In addition, Cutoff Finder tool available at http://molpath.charite.de/cutoff/ was used to investigate the influence of a single continuous variable on survival (*Budczies et al., 2012*). In the Cutoff Finder tool, the cutoff for dichotomization of a continuous variable was determined as the point with the most significant split by log-rank test, using *coxph* and *survfit* functions from the R package survival. Survival analysis of either *CASP8*-WT or *CASP8*-MT cases was performed using this method by dichotomizing gene expression or immune cell infiltration levels.

# RESULTS

## Genes involved in immune response are up-regulated in mutant-*CASP8* HNSCs

To investigate the significance of *CASP8* mutations in head and neck squamous cell carcinoma (HNSC), we performed differential gene expression analysis using RNA-seq data from HNSC cases with and without *CASP8* mutations. As reported previously, HNSCs carrying *CASP8* mutations occurred predominantly in sites within the oral cavity such as the cheek mucosa, floor of mouth, tongue, larynx, and overlapping sites of the lip, oral cavity, and pharynx (Table S1). In addition, since HPV-positive (Human Papillomavirus-positive) HNSCs constitute a molecularly distinct subtype; we examined the HPV status of the 55 mutant-*CASP8* HNSCs using data from *Chakravarthy et al. (2016)*. Based on this reported data, all 55 mutant-*CASP8* HNSCs were found to be HPV-negative. Since all the HNSC cases carrying *CASP8* mutations were HPV-negative, and were from specific sites within the oral cavity, HNSCs carrying wild-type-*CASP8* that were HPV-negative and also from these same sites were selected as controls for all subsequent analyses. A total of 424 HNSC cases of which 369 had wild-type-*CASP8* (*CASP8*-WT) and 55 had mutant-*CASP8* (*CASP8*-MT)

were thus selected (Fig. 1, see also Table S1). Of these, RNA-seq data was available for 354 *CASP8*-WT and 53 *CASP8*-MT cases.

Raw sequencing reads from *CASP8*-WT and *CASP8*-MT cases, obtained from HT-Seq files, were subjected to edgeR analysis for differential gene expression (Table S2). At FDR < 0.001, 186 genes were up-regulated in *CASP8*-MT with log2FC > 1.3 while 1,139 genes were down-regulated in *CASP8*-MT with log2FC < −1.3 (Fig. 2A). There was also a statistically significant 1.3-fold increase in the expression level of the *CASP8* gene perhaps to overcome the loss of function (Table S2, gene ESNG00000064012 in HNSC edgeR output).

To identify biological processes specifically enriched in the *CASP8*-WT or *CASP8*-MT cases, enrichment analysis was performed with the differentially expressed genes using tools available at the Gene Ontology (GO) Consortium. As seen in Fig. 2A, distinct processes were enriched in the *CASP8*-WT and *CASP8*-MT cases. For example, genes involved in the regulation of immune response (*p*-value = 3.30E−02) were enriched in *CASP8*-MT HNSCs while genes with roles in synaptic transmission (*p*-value = 2.85E−07), synaptic vesicle exocytosis (*p*-value = 3.90E−03), and muscle contraction (*p*-value = 2.29E−03) were the top three biological processes enriched in *CASP8*-WT HNSCs. Please refer to Table S3 for the full list.

We further analyzed the differential gene expression data using the Gene Set Enrichment Analysis (GSEA) tool. After generating a pre-ranked gene list based on logFC values from the edgeR analysis, we queried this list in the GSEA software using hallmark gene sets available at the Molecular Signatures Database. Several gene sets were enriched in upregulated or downregulated genes in *CASP8*-MT cases at FDR < 25%. Particularly, gene sets involved in immune regulation such as allograft rejection, complement, inflammatory response, interferon-$\alpha$ response, and interferon-$\gamma$ response, were specifically enriched in the *CASP8*-MT HNSCs, in sync with the GO results (Figs. 2B–2F and Table S4). The hallmark gene sets enriched in *CASP8*-WT HNSCs were epithelial-mesenchymal transition (EMT), myogenesis, and the KRAS pathway (Figs. 2G–2I and Table S4).

## Gene expression in the skins of epidermal *Caspase-8* knockout mice mirrors the expression pattern of mutant-*CASP8* HNSCs

Expression of an enzymatically inactive *Caspase-8* mutant or the deletion of wild-type *Caspase-8* in the mouse epidermis leads to chronic skin inflammation (*Kovalenko et al., 2009*; *Lee et al., 2009*). A microarray analysis performed by Kovalenko et al. to identify genes specifically up-regulated in the skin epidermis of *Casp-8*F/− *K5-Cre* (relative to *Casp-8*F/+ *K5-Cre* epidermis) mice revealed increased expression of several immune-regulatory and inflammatory genes including several cytokines. Using the human orthologs of these up-regulated genes (Table S5), we again queried the pre-ranked gene list with the GSEA tool. As seen in Fig. 2J, genes highly expressed in the *Casp-8*F/− *K5-Cre* mouse skins were also significantly enriched in *CASP8*-MT HNSCs (as opposed to their wild-type counterparts), indicating that the inactivation of *CASP8* leads to the up-regulation of a similar set of genes in both mouse and human epidermal tissues.

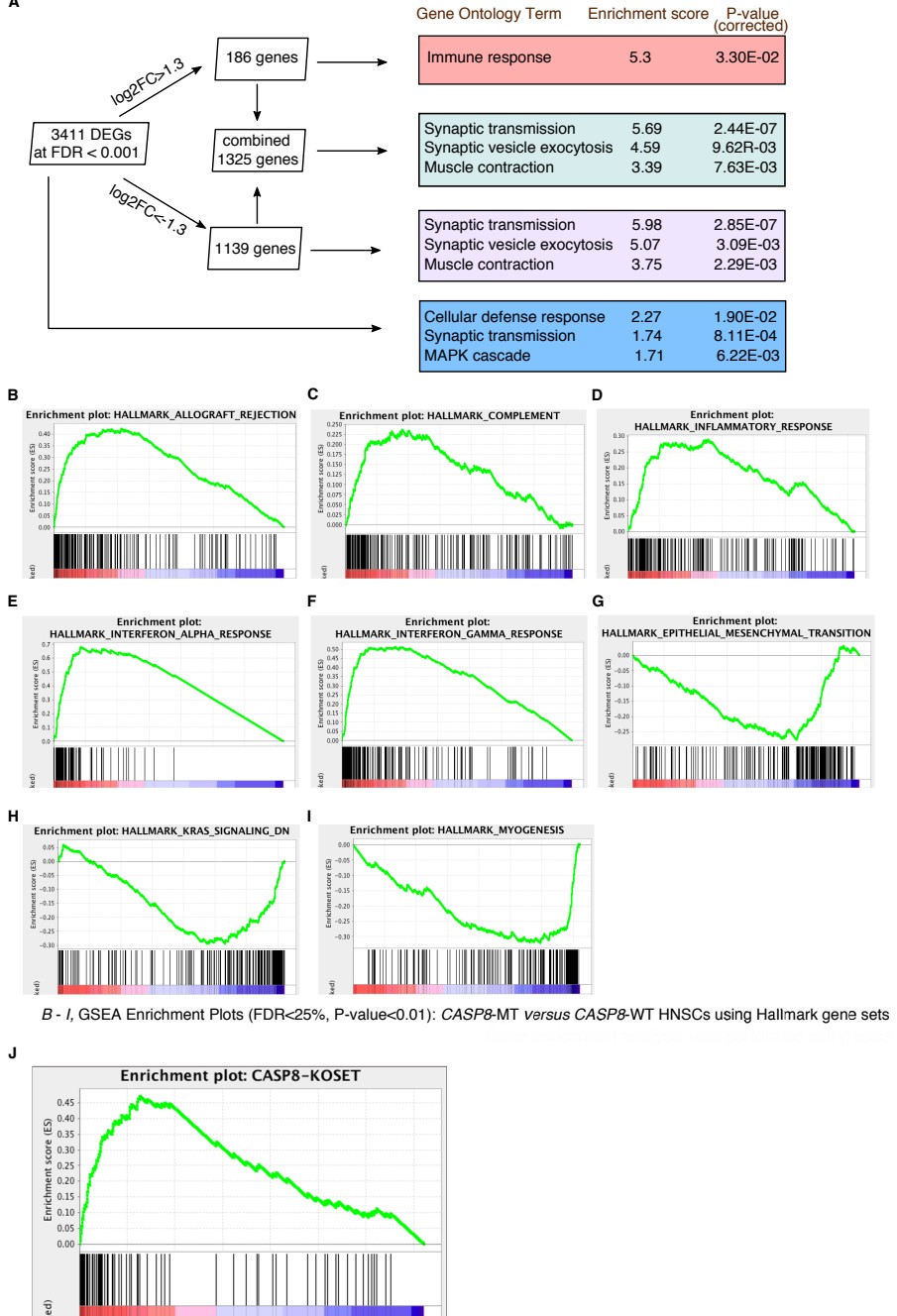

B - I, GSEA Enrichment Plots (FDR<25%, P-value<0.01): *CASP8*-MT *versus CASP8*-WT HNSCs using Hallmark gene sets

GSEA Enrichment Plot: *CASP8*-MT *versus CASP8*-WT
using *CASP8*-KOSET (FDR<25%, P-value<0.01)

**Figure 2** **Gene enrichment analyses reveal a prominent immune signature in *CASP8*-MT HNSCs.**
Gene enrichment analysis was performed using tools available at the Gene Ontology Consortium (A), as
well as using the Gene Set Enrichment Analysis tool (B–J). (A) Enrichment analysis was performed using
genes with FDR < 0.001 and/or showing log2FC greater than 1.3 or less than −1.3. The top three gene on-
tology terms, based on enrichment scores, among the PANTHER GO-Slim Biological Processes signifi-
cantly enriched in these gene lists are indicated along with Bonferroni-corrected *P*-values. 

**Figure 2 (…continued)**
(B–I) GSEA was performed using a pre-ranked list generated using log2FC values from the edgeR analysis. GSEA Hallmark gene sets enriched in *CASP8*-MT HNSCs (B–F) or *CASP8*-WT HNSCs (G–I) with FDR < 25%, *P*-value < 0.01, and showing enrichment at the top or bottom of the list are shown. (J) Enrichment plot of a GSEA performed with the same pre-ranked list that was analysed in (B–I) and a gene set of human orthologs of the genes up regulated in the skin epidermis of *Casp-8*^F/−^ *K5-Cre* mice (*CASP8*-KOSET) is shown (FDR < 25%, *P*-value < 0.01).

## Enrichment of immune response gene sets correlates with increased infiltration of specific immune cell types in mutant-*CASP8* HNSCs

Gene sets involved in immune response were specifically enriched in *CASP8*-MT HNSCs. HPV-positive HNSCs, a subset of HNSCs, also display high immune cell infiltration, as compared to HPV-negative HNSCs (*Chakravarthy et al., 2016*; *Mandal et al., 2016*; *Nguyen et al., 2016*; *Russell et al., 2013*). We investigated if the enrichment of immune response genes in *CASP8*-MT HNSCs was correlated with increased infiltration of immune cells, and if it was comparable to the immune cell infiltration levels in HPV-positive HNSCs. Immune cell infiltration levels in three subsets of HNSCs; *CASP8*-WT, *CASP8*-MT (both HPV-negative), and HPV-positive (which is *CASP8*-WT), were compared using the Wilcoxon test; the comparisons were: (1) *CASP8*-WT and *CASP8*-MT (2) *CASP8*-WT and HPV-positive *CASP8*-WT, and (3) *CASP8*-MT and HPV-positive *CASP8*-WT. We checked if there was a difference in the numbers/types of immune cell infiltrates between these three subsets of HNSCs using the data available at Tumor IMmune Estimation Resource (TIMER) (*Li et al., 2017*), a comprehensive resource for immune cell infiltration of TCGA tumors.

Consistent with the GSEA results, *CASP8*-MT cases showed significantly higher infiltration of $CD8^+$ T cells, neutrophils, and dendritic cells as compared to *CASP8*-WT cases ($p$-values < 0.0005), suggesting that the immune response to the tumor in WT and MT cases was different (Figs. 3A–3C). Also, in agreement with previous reports, HPV-positive HNSCs had significantly higher infiltration of all immune cell types as compared to the *CASP8*-WT HNSCs. A comparison of immune cell infiltration levels in HPV-positive and *CASP8*-MT HNSCs showed that the extent of infiltration of $CD8^+$ T cells, neutrophils, and dendritic cells (Figs. 3A–3C) in these two subsets was also similar. However, HPV-positive HNSCs had higher infiltration of $CD4^+$ T cells and B cells, compared to the other two subsets (Figs. 3D, 3E).

## The "immune signature" of mutant-*CASP8* HNSCs does not correlate to improved overall survival

High levels of immune cell infiltration in HPV-positive cases correlates with better survival in HPV-positive HNSC cases (*Nguyen et al., 2016*; *Russell et al., 2013*). To investigate if a similar effect could be observed in the survival of HNSC patients with and without *CASP8* mutation, Kaplan–Meier analysis was performed on the *CASP8*-WT and *CASP8*-MT cases (filtered as per the schema in Fig. 1). There was no significant difference in the survival of patients with and without *CASP8* mutations ($p$-value = 0.16, Fig. 4A), indicating that high levels of immune cell infiltration may not necessarily corelate with better survival in HNSCs.

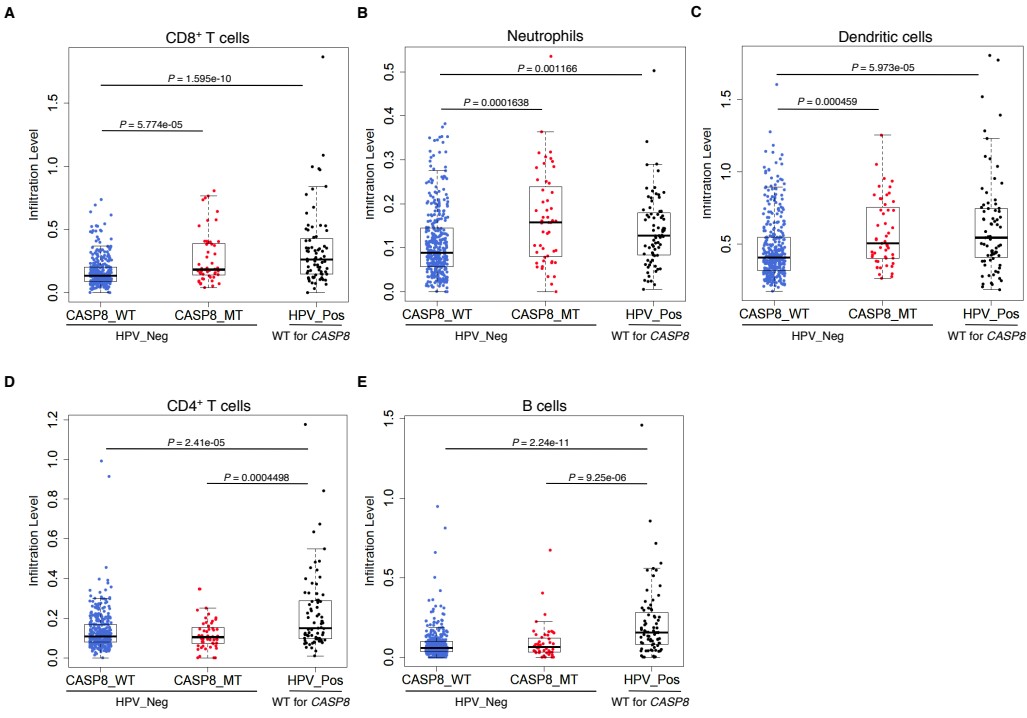

**Figure 3** *CASP8*-MT HNSCs have higher numbers of certain types of infiltrating immune cells compared to *CASP8*-WT HNSCs. Immune cell infiltration levels in *CASP8*-WT (blue-filled circles), *CASP8*-MT (red-filled circles) (both HPV-negative), and HPV-positive (black-filled circles) HNSCs were compared using the immune cell infiltration data available at TIMER. Boxplots showing the levels of CD8⁺ T cells, neutrophils, and dendritic cells (A–C), as well as CD4⁺ T cells and B cells (D, E) in the three HNSC subsets are displayed. Significance testing was performed using the unpaired two-sided Wilcoxon test. All comparisons with *P*-value < 0.005 were considered significant and are indicated in the plots.

    The effect of genes from pathways enriched either in *CASP8*-WT or *CASP8*-MT tumors (listed in Table S4) on survival was then investigated using the Cox proportional hazards model. Four genes from pathways enriched in *CASP8*-MT HNSCs; *PRF1*, *CXCR6*, *CD3D*, and *GZMB*, reduced the hazard ratio significantly in *CASP8*-WT cases, at $p < 0.05$ (Table S6). We also performed the survival analysis using Cutoff Finder to investigate the effect of the expression of individual genes on the survival of *CASP8*-WT and *CASP8*-MT cases. Increased expression of all these four genes was associated with higher overall survival in *CASP8*-WT (at $p < 0.05$). In contrast, in *CASP8*-MT cases, such association was seen only with *GZMB* expression levels (Figs. 4B, 4C and Fig. S1). Similarly, higher CD8⁺ T cell estimates (from TIMER) was also significantly associated with better survival in *CASP8*-WT but not in *CASP8*-MT HNSCs (Figs. 4D, 4E).

    Since CASP8 is a negative regulator of the necroptotic pathway (*Günther et al., 2011*; *Weinlich et al., 2013*), we also investigated the effect of expression levels of genes involved in necroptosis on survival. Higher expression of *RIPK1*, *RIPK3,* and *MLKL* was associated with higher overall survival in *CASP8*-WT but not in *CASP8*-MT cases (Fig. S2). Additional factors that influenced survival are shown in Table S6.

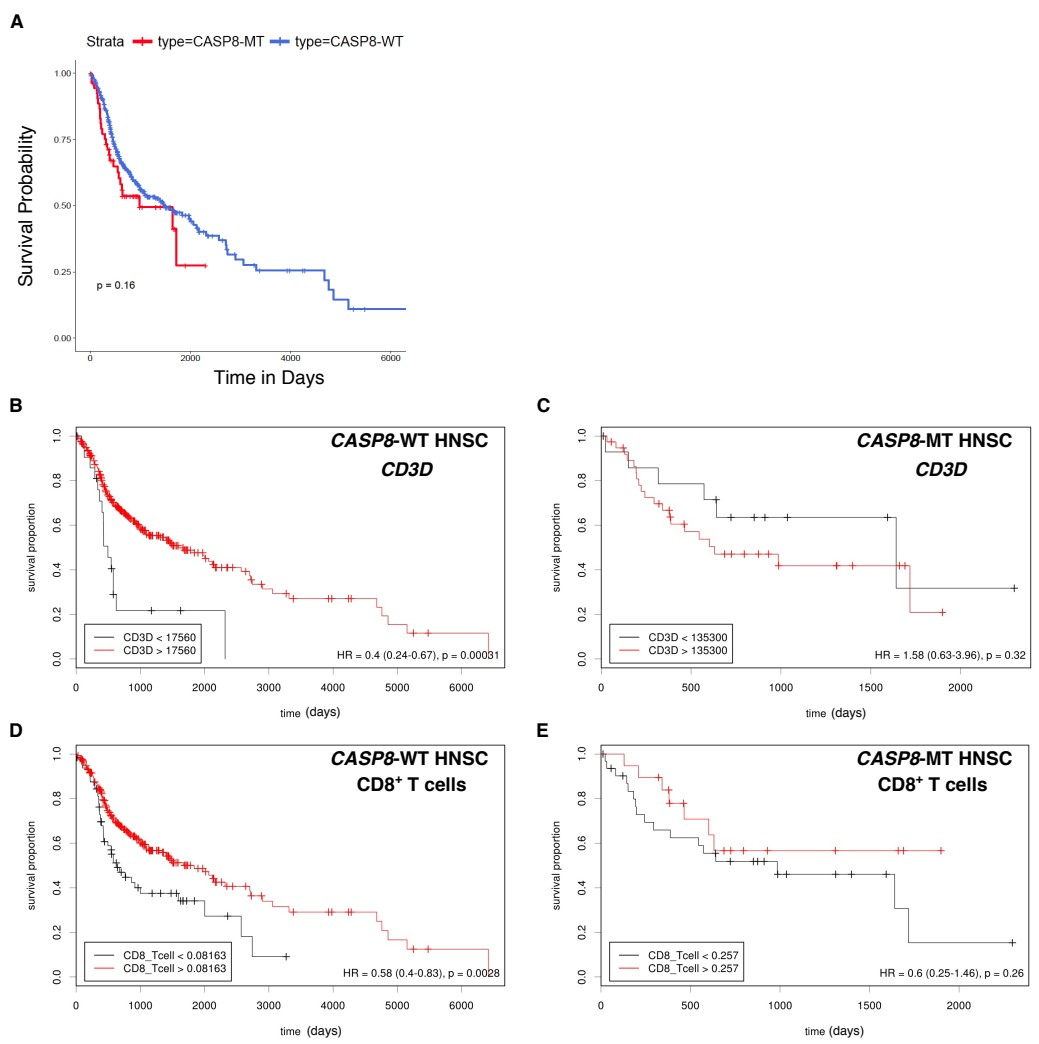

**Figure 4** **Survival analysis indicates lack of a survival advantage in *CASP8*-MT HNSCs in spite of their immune signature.** (A) Kaplan-Meier plots showing the survival probability of patients with *CASP8*-WT or *CASP8*-MT HNSC tumors (filtered as per the schema in Fig. 1). Log-rank test was used to compare the two curves and the log-rank *P*-value is indicated. (B–E) Survival plots generated using the Cutoff Finder tool showing the influence of the expression levels of *CD3D* and the levels of CD8+ T cells on overall survival in *CASP8*-WT (B, D) and *CASP8*-MT (C, E) cases. Gene expression data was obtained from FPKM-UQ files at TCGA and immune cell infiltration data was obtained from TIMER.

## Mutant-*CASP8* UCECs exhibit an immune signature similar to mutant-*CASP8* HNSCs

We then investigated if this effect seen in *CASP8*-MT HNSCs was broadly applicable across other cancers carrying *CASP8* mutations. On searching the Genomic Data Commons, we found that *CASP8* was recurrently mutated in about 10% of (UCEC) cases. From a total of 560 UCEC cases, RNA-seq data was available for 476 *CASP8*-WT and 56 *CASP8*-MT cases. HTSeq files containing raw sequencing reads from these two groups were subjected to edgeR analysis for differential gene expression and further analyzed using the GSEA
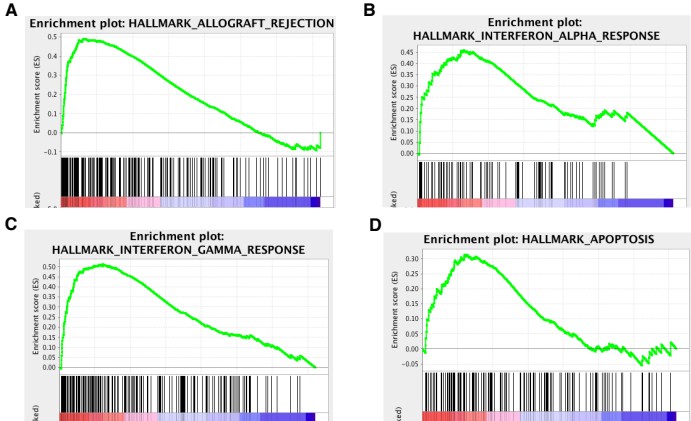

*A - D,* GSEA Enrichment Plots: *CASP8*-MT *versus CASP8*-WT UCECs using Hallmark gene sets

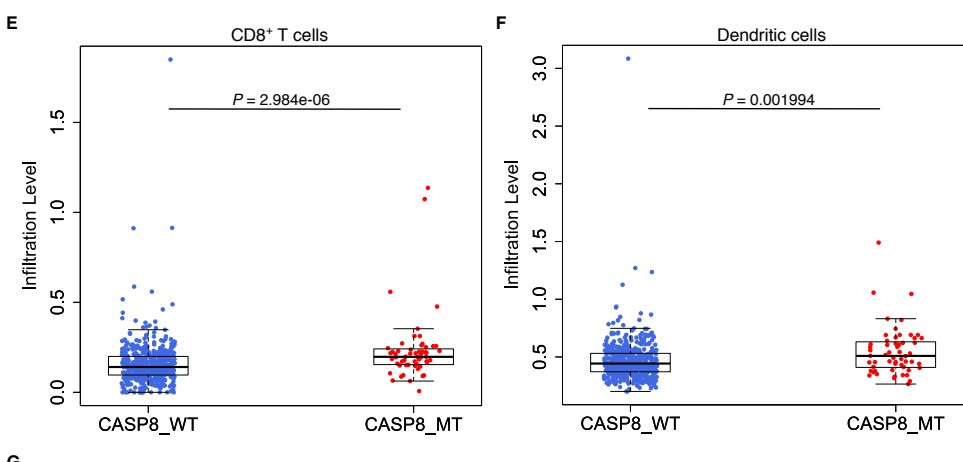

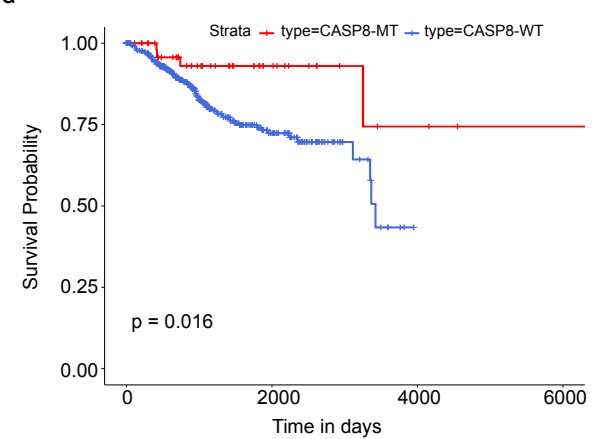

**Figure 5** ***CASP8*-MT UCECs display an immune gene signature, have higher numbers of certain types of infiltrating immune cells, and survive better than *CASP8*-WT UCECs.** (A–D) GSEA was performed using a pre-ranked list generated using log2FC values from the edgeR analysis. Some GSEA Hallmark gene sets enriched in *CASP8*-MT UCECs (A–D) are shown. (E, F) Immune cell infiltration levels in *CASP8*-WT (blue-filled circles) and *CASP8*-MT (red-filled circles) UCECs were compared using the immune cell infiltration data available at TIMER. Boxplots showing the levels of CD8+ T cells and dendritic cells in the two UCEC groups are displayed. 

**Figure 5 (…continued)**
Significance testing was performed using the unpaired two-sided Wilcoxon test. All comparisons with *P*-value < 0.005 were considered significant and are indicated in the plots. (G) Kaplan-Meier plots showing the survival probability of patients with *CASP8*-WT or *CASP8*- MT UCEC tumors. Log-rank test was used to compare the two curves and the log-rank *P*-value is indicated.

tool. Several gene sets involved in immune regulation were specifically enriched in the *CASP8*-MT UCECs. Notably, categories such as allograft rejection, interferon-$\alpha$ response, and interferon-$\gamma$ response, were enriched in the *CASP8*-MT UCECs similar to the HNSC results (Figs. 5A–5C and 2B–2F). The genes that contributed to core enrichment in these gene sets in *CASP8*-MT UCECs also contributed to core enrichment of the same gene sets in *CASP8*-MT HNSCs, indicating that similar immune response genes were upregulated in the two carcinomas (Table S7). However, unlike HNSCs, the gene set for inflammatory response did not show any enrichment in *CASP8*-MT UCECs. *CASP8*-MT UCECs were additionally enriched for genes involved in apoptosis. Notably, this was not observed in the *CASP8*-MT HNSCs (Figs. 2B–2I and 5D).

### High levels of IL33 and neutrophil infiltration are observed in mutant-*CASP8* HNSCs but not in mutant-*CASP8* UCECs

Using TIMER, we then checked the levels of infiltrating immune cells in the *CASP8*-WT and *CASP8*-MT UCECs. Consistent with the GSEA results, *CASP8*-MT UCEC cases showed significantly higher infiltration of $CD8^+$ T cells and dendritic cells as compared to *CASP8*-WT cases (*p*-values < 0.005). However, in contrast to the HNSC data, the levels of neutrophils were not significantly higher in the *CASP8*-MT UCEC group (Figs. 5E, 5F, see also Fig. 3B). We then investigated if differences in the levels of neutrophil-active chemokines could potentially explain this observation (Sadik, Kim & Luster, 2011). From the edgeR differential expression data comparing the *CASP8*-MT and *CASP8*-WT groups in HNSC and UCEC, we obtained the fold change values and statistical significance of different chemokines known to attract neutrophils (Table S8). Interestingly, the cytokine IL33 was significantly up regulated (1.8 fold, FDR < 0.001) in *CASP8*-MT HNSCs but not in *CASP8*-MT UCECs.

Next, we performed Kaplan–Meier analysis on *CASP8*-WT and *CASP8*-MT UCEC cases. In contrast to the HNSC survival data, there was a difference in the survival of UCEC cases with and without *CASP8* mutations, with cases harboring *CASP8* mutations reporting better overall survival (*p*-value = 0.019, Fig. 5G).

## DISCUSSION

Here, we report a distinct class of carcinomas that have mutated *CASP8*. Using bioinformatics approaches to mine the TCGA data, we identified high expression of immune response-related genes (listed in Table S7) combined with high infiltration of $CD8^+$ T cells and dendritic cells as a prominent shared immune signature in *CASP8*-MT carcinomas. In the first part of our analyses, we investigated the implications of the enrichment of this immune signature across different HNSC subtypes. Subsequently, in the second part, we investigated the correlation between immune signature and survival in
two carcinomas having a significant number of cases with *CASP8* mutations, HNSC and UCEC. Our analyses showed that despite similarities in the enrichment of gene sets, these carcinomas exhibited varying correlations of immune signature with survival. Our studies indicated that tissue-specific differences, such as the levels of infiltrating neutrophils and the cytokine IL33, could be responsible for the varying correlation of immune signature with survival.

Multiple studies have reported that HPV-positive HNSCs display a strong immune signature and high infiltration of immune cells that correlates with better survival (*Nguyen et al., 2016*; *Russell et al., 2013*). In contrast, our studies show that the enrichment of immune response genes and infiltration of immune cells seen in *CASP8*-MT HNSCs does not appear to correlate with improved prognosis. In fact, *CASP8* mutation leads to the loss of a survival advantage that is observed in HNSC patients with wild-type *CASP8* tumors under certain conditions. For example, higher expression levels of genes such as *PRF1*, *CD3D*, and *CXCR6* are associated with better survival in *CASP8*-WT but not in *CASP8*-MT. It is possible that the higher expression of these genes results in higher extent of apoptosis leading to survival advantage. This perhaps does not take place in *CASP8*-MT, leading to the loss of survival advantage from higher expression of these genes. These results argue that a tumor microenvironment with high infiltration of immune cells does not necessarily provide a survival benefit in HNSCs. However, it is important to note that while the sample numbers of *CASP8*-MT cases are significant ($n = 55$), it is lesser than the number of *CASP8*-WT cases ($n = 369$). This may influence $p$-values, and it will be necessary to confirm these findings with greater numbers of *CASP8*-MT cases once more data becomes available.

We can think of at least two potential scenarios to explain the increased immune cell infiltration observed in *CASP8*-MT tumors. (a) Unregulated inflammatory and wound healing response: As mentioned earlier, loss of *Caspase-8* in the mouse epidermis leads to chronic inflammation (*Kovalenko et al., 2009*). The infiltration of immune cells in mucosa lacking *CASP8* accompanied by the enrichment of immune-associated gene sets is highly reminiscent of this phenotype. It has also been proposed that the loss of *Caspase-8* in the mouse skin epidermis simulates a wound healing response (*Lee et al., 2009*). Both scenarios involve a gamut of immune cell types and secreted cytokine factors, leading to immune cell infiltration. It should however be noted that although similar gene sets are enriched in mouse skins lacking *Caspase-8* and in *CASP8*-MT tumors, the types of immune cell infiltrates in the two are different. (b) Necroptosis: More recently, several studies have revealed a role for Caspase-8 as an inhibitor of necroptosis, a highly pro-inflammatory mode of cell death (*Pasparakis & Vandenabeele, 2015*; *Feltham, Vince & Lawlor, 2017*). In intestinal epithelia, the loss of *Caspase-8* promoted necroptosis through the activation of RIP kinases and MLKL (*Günther et al., 2011*; *Weinlich et al., 2013*). A similar scenario could be occurring in *CASP8*-MT tumors leading to the expression of pro-inflammatory genes and the infiltration of immune cells.

Why doesn't the increased number of immune cells translate into improved prognosis in *CASP8*-MT HNSC tumors? Since CASP8 is an important mediator of the extrinsic apoptotic pathway, *CASP8*-MT tumors may have greater resistance to Fas- or DR5- mediated cell

death pathways, which are typically employed by CD8[+] T cells and Natural Killer cells to target infected/tumor cells (*Li et al., 2014*; *Rooney et al., 2015*). The survival analysis carried out in this study showed that *CASP8*-WT HNSC patients with higher expression of genes involved in T-cell mediated cytotoxicity had better survival. Importantly, this advantage was not seen in *CASP8*-MT patients.

Several studies have reported that high neutrophil numbers and an elevated neutrophil/lymphocyte ratio portended poorer prognosis in OSCC (*Mahalakshmi et al., 2018*; *Glogauer et al., 2015*). Thus, it is possible that elevated levels of neutrophil infiltration seen in *CASP8*-MT HNSC cases could be one of several events contributing to the poorer prognosis of *CASP8*-MT HNSCs. IL33, a cytokine and an alarmin linked to necroptosis may represent a possible mechanism for neutrophil recruitment in these cases (*Alves-Filho et al., 2010*; *Hueber et al., 2011*). High IL33 levels are also associated with poor prognosis in HNSCs (*Chen et al., 2013*). In addition, the pro-inflammatory environment generated during necroptosis may hold other advantages for the survival of *CASP8*-MT HNSCs. Necroptosis, IL33 levels, and neutrophil infiltration together or through independent mechanisms could be leading to a pro-tumor environment. Thus, promoting necroptosis may not necessarily translate into better survival for HNSC patients with apoptosis-resistant tumors.

Another reason for the lack of survival advantage in *CASP8*-MT HNSCs could be the composition of tumor-infiltrating immune cells in these tumors. For instance, HPV-positive tumors had higher levels of B cells and CD4[+] T cells as compared to *CASP8*-MT tumors. It is likely that in addition to cytotoxic T cells, B cells and CD4[+] T cells are required to mediate an immune response essential for tumor cell death, possibly for tumor antigen presentation or cytokine secretion.

A comparison of *CASP8*-MT HNSCs and *CASP8*-MT UCECs highlighted similarities and differences between the two carcinomas. Both *CASP8*-MT HNSCs and *CASP8*-MT UCECs showed an enrichment of gene sets involved in immune response such as interferon $\alpha$ response, interferon $\gamma$ response, and allograft rejection. Most of the genes contributing to core enrichment in these gene sets in UCECs also contributed to core enrichment of these gene sets in HNSCs. Moreover, both these carcinomas showed high infiltration of CD8[+] T cells and dendritic cells but not B cells or CD4[+] T cells. *CASP8* mutation thus led to a similar immune response in both HNSCs and UCECs. This shared immune signature, however, did not correlate with a specific survival outcome. Notably, *CASP8*-MT UCECs showed a significant survival advantage over *CASP8*-WT UCECs, unlike its HNSC counterpart. While we do not yet know the causal reason(s), the differences *per se* may be worth noting and could be responsible for this advantage. For instance, in contrast to *CASP8*-MT HNSCs, the gene set for inflammatory response was not enriched but the gene set for apoptosis was enriched in *CASP8*-MT UCECs. There was also no increased infiltration of neutrophils or transcriptional upregulation of IL33 in *CASP8*-MT UCECs. The up-regulation of apoptotic pathways together with the lack of enrichment of an inflammation-associated gene set that is typical of necroptosis perhaps indicates that apoptosis, rather than necroptosis, is the predominant mode of programmed cell death in *CASP8*-MT UCECs. This lack of inflammation may also be responsible for the lack

of neutrophil infiltration in *CASP8*-MT UCECs since neutrophil chemoattractants, such as IL33, may not be released during apoptosis but is perhaps released during the highly inflammatory process of necroptosis, in turn leading to neutrophil infiltration.

It is also possible that necroptosis is initiated in *CASP8*-MT UCECs but the accompanying IL33 up-regulation and/or neutrophil infiltration seen in HNSCs does not take place due to tissue-specific differences. Under such conditions, apoptosis and necroptosis together could provide the survival advantage that is observed in *CASP8*-MT UCECs. Thus, in contrast to HNSCs, Caspase-8 pathway can be explored to identify potential drug targets in UCECs.

## CONCLUSIONS

In this *in silico* study, we explore the implications of *CASP8* mutations that have been identified across carcinomas through large-scale genomic studies. Our studies show that *CASP8*-mutated carcinomas display a shared immune signature. However, the consequences of this immune signature vary with *CASP8*- MT UCECs showing better survival while *CASP8*-MT HNSC cases do not have any survival advantage. Our analyses further suggest that neutrophil numbers and IL33 levels could be potential factors affecting the survival of mutant-*CASP8* carcinomas. Broadly, our study highlights the need to further investigate the interaction between pathways of programmed cell death, immune response, and survival in carcinomas. Such studies could open a new window for therapeutic intervention in *CASP8*-mutated carcinomas.

## ACKNOWLEDGEMENTS

The results shown here are based upon data generated by the TCGA Research Network: https://cancergenome.nih.gov/. We thank patients who donated samples to the TCGA and consented to share the resulting data. We also wish to thank TCGA for the unrestricted access provided to the data used in this work. We would like to thank Amrendra Mishra (Hannover Biomedical Research School), Urvashi Bahadur (Strand Life Sciences), and Colin Jamora (inStem) for their helpful comments on this manuscript. SS thanks S Ramaswamy (inStem) for his support.

### Funding

Subhashini Sadasivam's work was supported by inStem core funds. The publication of this article was made possible through a grant (BT/PR17576/MED/30/1690/2016) from the Department of Biotechnology, Government of India. The funders had no role in study design, data collection and analysis, decision to publish, or preparation of the manuscript.

### Grant Disclosures

The following grant information was disclosed by the authors:
inStem core.
Department of Biotechnology, Government of India: BT/PR17576/MED/30/1690/2016.

## Competing Interests

Both the authors are co-founders of DeepSeeq Bioinformatics. Yashoda Ghanekar is CEO and Subhashini Sadasivam is CSO at DeepSeeq Bioinformatics. The authors declare there are no competing interests.

## Author Contributions

- Yashoda Ghanekar and Subhashini Sadasivam conceived and designed the experiments, performed the experiments, analyzed the data, contributed reagents/materials/analysis tools, prepared figures and/or tables, authored or reviewed drafts of the paper, approved the final draft.

## Data Availability

All data that was generated during this study are available in this article and the Supplemental Files. The results shown here are based upon data generated by the TCGA Research Network: https://portal.gdc.cancer.gov/projects/TCGA-HNSC.

## Supplemental Information

Supplemental information for this article can be found online at http://dx.doi.org/10.7717/peerj.6402#supplemental-information.

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
