# Peer review of "In silico analysis reveals a shared immune signature in CASP8-mutated carcinomas with varying correlations to prognosis"

_PeerJ, doi:10.7717/peerj.6402_

## Round 0.1 · original submission · Major Revisions

Dear Dr. Subhashini,

Thank you for the submission. It has been reviewed by two reviewers. Both of them recognized the significance of the paper. Please address the technical issues raised by the reviewers and clarify some confusions to improve the manuscript.

Best Regards,

Xiangqin Cui

Reviewer 1 ·

Basic reporting

1. Some citations are missing. Line 81, 82, 335.
2. Line 164, please be specific about what “data” you downloaded from TIMER. Data is too general.
3. Line 243-264, the description is quite chaotic here. The authors compare CASP8-MT to CASP8-WT and HPV-positive, which is unclear at all. By “CASP8-WT”, I guess you mean CASP8-WT with HPV-negative. By “HPV-positive”, I guess you mean CASP8-WT with HPV-positive, right? Please write these very clear so that the reader won’t confuse. Revise Figure 3a accordingly as well.
4. Line 292 and 293 should be in the introduction. Otherwise it also seems confusing why you choose UCEC instead of other cancer types.
5. Line 294-295 replicates the previous content. Should be removed.

Experimental design

1. It seems a bit mis-match between your title and your content. I tried very hard to figure out what this “shared immune signature” is, which seems to be the most important meessage in your title. However, it is not clear at all what this “shared immune signature” is by reading the main manuscript. Based on the abstract, it seems this immune signature is high neutrophil numbers (but the sixth section of results showed this feature is not shared in UCEC). Based on the fourth section of results part, it seems this immune signature is these four genes from pathways enriched in CASP8-MT HNSC. Furthermore, based on the fifth section in results, it seems this immune signature is these enriched categories such as allograft rejection.
2. A big chunk of differential gene expression analysis results seems to be missing. In the Materials and Methods section, you describe that the RNA-seq data from CASP8-MT and CASP8-WT are compared through differential gene expression analysis. However, the results section only present the CASP9-WT who are HPV-negative versus CASP9-MT. What about the comparison results of the whole group of CASP9-WT versus CASP9-MT regardless of their HPV status? Why you only exclude HPV-positive patient from this analysis? I understand that CASP9-MT patients are all HPV-negative, but HPV is just a sub-feature. CASP9-MT or WT should be the main feature regardless of their HPV status.
3. In the introduction section, the authors state that among all the genes, “TP53 was the most significant recurrently mutated genes”. But instead, they decide to focus on another gene “CASP8”. What is the motivation? Why they do not want to study TP53 here? What is so special about “CASP8”?
4. You grouped CASP8 high mutations and mediate mutations to one group. What if you consider them differently? Will the patients have more impact with severe mutations?

Validity of the findings

1. In the Gene Onotology and GSEA part of Method section, the authors identified 37095 genes. But as I recall, there are a total of 19000-20000 genes in homo sapiens. I wonder whether they mapped this correctly.
2. Again in this GSEA part, they seems to use a set of genes from mouse study. I am not quite confident in such comparisons. Unless the authors can provide evidence that this comparison is a common practice, I tend to suggest that the authors should find relevant gene sets from human study. Related to this, I want to recommend three other GO and pathway tool: EnrichR (http://amp.pharm.mssm.edu/Enrichr/), GREAT (http://great.stanford.edu/public/html/) and DAVID (https://david.ncifcrf.gov/) which offers GO and pathway analysis, even GSEA analysis in one stop. They may be more reliable than basing on a single gene set.

Reviewer 2 ·

Basic reporting

This paper is clearly written and organized. Sufficient introduction and background is provided. Figures are of good quality and addressed the relevant questions. The resources for publicly available data used in this study is clearly referenced.

Experimental design

no comment

Validity of the findings

no comment

Additional comments

1. About the motivation of this study, the authors stated that CASP8 in one of the highly mutated genes in HNSC. My question if why chooses to study CASP8 instead of other highly mutated genes? Need more detailed explanations and discussion here.


2. The authors used pre-ranked GSEA instead standard GSEA to interpret gene expression differences between CASP8-WT and CASP8-MT. However, it is well known that pre-ranked GSEA inflates p-values due to the intercorrelation of genes, and it is usually only used when there is no better alternative approaches, for example, RNA-seq data with no or few replicates. But apparently in TCGA, the number of replicates is not an issue. Therefore, my suggestion is to use the standard GSEA instead of pre-ranked GSEA, and to see if still get the similar results.


3. In Figure 4B and Figure S1, the authors compared the effects of immune signature gene expression on survival in CASP8-MT and CASP8-WT samples, and made the interpretation that higher expression of immune signature genes (and CD8+) is only correlated with better survival in CASP8-WT but not in CASP8-MT. My concern is this is a rather strong conclusion, since there is a trend of better survival in CASP8-MT and the p-value is not significant enough probably due to the much small sample size camped to CASP8-WT.

4. The authors did a great job at discussing the potential reasons why CASP8-MT UCECs showed a significantly better survival while the CASP8-MT HNSCs did not. However, the conclusion that this difference might be “attributed to differences in neutrophil infiltration and/or IL33 levels” is way too strong. I would suggest either provide more supporting evidence (which is probably very difficult) or change the conclusions.

---

## Round 0.2 · accepted · Accept

Congratulations! Your manuscript is accepted.

# Reviewer 1 ·

Basic reporting

no comment

Experimental design

no comment

Validity of the findings

no comment

Additional comments

no comment

Reviewer 2 ·

Basic reporting

The authors have addressed my concerns and questions in the revision.

Experimental design

no comment

Validity of the findings

no comment

Additional comments

no comment